# Equilibrium and Kinetic Study of Anionic and Cationic Pollutants Remediation by Limestone–Chitosan–Alginate Nanocomposite from Aqueous Solution

**DOI:** 10.3390/molecules26092586

**Published:** 2021-04-29

**Authors:** Inas A. Ahmed, Ahmed H. Ragab, Mohamed A. Habila, Taghrid S. Alomar, Enas H. Aljuhani

**Affiliations:** 1Department of Chemistry, Faculty of Science, King Khalid University, Abha 62224, Saudi Arabia; eaahmed@kku.edu.sa; 2Chemistry Department, College of Science, King Saud University, Riyadh 11451, Saudi Arabia; Mhabila@ksu.edu.sa; 3Department of Chemistry, College of Science, Princess Nourah Bint Abdulrahman University, Riyadh 11671, Saudi Arabia; 4Department of Chemistry, Collage of Applied Science, Umm Al-Qura University, Mecca 21955, Saudi Arabia; ehgohani@uqu.edu.sa

**Keywords:** nanolimestone, alginate, brilliant green, Congo red, adsorption

## Abstract

In this work, low-cost and readily available limestone was converted into nanolimestone chitosan and mixed with alginate powder and precipitate to form a triple nanocomposite, namely limestone—chitosan–alginate (NLS/Cs/Alg.), which was used as an adsorbent for the removal of brilliant green (BG) and Congo red (CR) dyes in aqueous solutions. The adsorption studies were conducted under varying parameters, including contact time, temperature, concentration, and pH. The NLS/Cs/Alg. was characterized by SEM, FTIR, BET, and TEM techniques. The SEM images revealed that the NLS/Cs/Alg. surface structure had interconnected pores, which could easily trap the pollutants. The BET analysis established the surface area to be 20.45 m^2^/g. The recorded maximum experimental adsorption capacities were 2250 and 2020 mg/g for CR and BG, respectively. The adsorption processes had a good fit to the kinetic pseudo second order, which suggests that the removal mechanism was controlled by physical adsorption. The CR and BG equilibrium data had a good fit for the Freundlich isotherm, suggesting that adsorption processes occurred on the heterogeneous surface with a multilayer formation on the NLS/Cs/Alg. at equilibrium. The enthalpy change (ΔH^0^) was 37.7 KJ mol^−1^ for CR and 8.71 KJ mol^−1^ for BG, while the entropy change (ΔS^0^) was 89.1 J K^−1^ mol^−1^ for CR and 79.1 J K^−1^ mol^−1^ BG, indicating that the adsorption process was endothermic and spontaneous in nature.

## 1. Introduction

In recent decades, rapid industrialization and urbanization have caused a disturbing global increase in water pollution. Removing dyed water is an absolutely crucial matter. Among many substances that pollute lakes and rivers, those of unnatural texture are very common [1]. About 1.0 × 10^5^ synthetic dyes and pigments, and more than 7.0 × 10^5^ tons of them, are consumed by industry annually [2]. Incomplete dying results in the discharge of polluted wastewater that can contaminate groundwater and surface water resources. Approximately 15% of organic dyes are released into the environment, causing environmental pollution [3]. The toxicity of effluents from textile industries is characterized by multiple variations in BOD, turbidity, color, COD, pH, odor, and salinity [4]. Azo dyes, such as Congo red, are highly toxic, carcinogenic, and mutagenic. They are hazardous to the respiratory system because of their vapor, and they are dangerous in cases of skin contact, eye contact, and swallowing [5,6,7,8,9,10]. During their degradation, carbon dioxide, nitrogen oxide, and sulfur oxides may form. Therefore, there is an urgent need to separate these dyes from wastewater [11,12,13,14,15,16,17].

Due to the potential toxicity of dyes in natural water ecosystems, the presence of dyes in surface water and groundwater represents a serious problem. The most recommended and applied technique to remove toxic dye is adsorption. Some cost-effective organic and inorganic adsorbents include peat [18], grapefruit peel [19], rice husks [20], peanut husks [21], banana peel [22], wood apple shells [23], orange peel [24], and mushroom waste [25]. Carbon [26] and polymers [27,28] are also used.

In this study, we made an environmentally friendly nanocomposite from sedimentary solid rock for the efficient removal of Congo red (CR) and brilliant green (BG) dyes from contaminated water. The limestone–chitosan–alginate (NLS/Cs/Alg.) nanocomposite adsorption capacity was examined using many techniques. The authors sought to quantify the effects of pH, dye concentration, and adsorbent dose on contact time. Sorption models were used to study the adsorption mechanism and the reaction order. The enthalpy, the Gibbs free energy, and the entropy of reaction were studied.

## 2. Results and Discussion

### 2.1. Structural and Surface Characterization of the Adsorbent

In order to identify different functional groups on the surface of the sorbent responsible for the binding of CR and BG dyes, FTIR spectral analysis was necessary. In Figure 1a, the band at 3449 cm^−1^ may be related to the (OH) groups present in the alginate, while the band at 1420 cm^−1^ may be due to the calcium ions in the nanocompound. The FTIR spectrum of the CR or BG with the NLS/Cs/Alg. nanocomposite showed some interesting features. In Figure 1b, a feature at 1799 cm^−1^ could be linked to the amide group stretching vibrations, while a more complex feature observed at 3499 cm^−1^ was due to -NH_2_ groups [29]. In Figure 1c, the key features here are a UV absorption band at around 1614 cm^−1^, probably due to the oxime group present in the brilliant green dye, and stretching bands at 874 cm^−1^ may be due to SO_3_^−^ groups [30].

#### 2.1.1. SEM Study

SEM images revealed that the adsorbent’s surface morphology had unique characteristics. Figure 2A–C shows examples of SEM images of CR and BG dyed on the NLS/Cs/Alg. polymers. The nanocomposites had varied surface shapes (before and after dye adsorption). The SEM images showed that before adsorption, the nanocomposite surface was rough and made of holes and small openings, whereas, after adsorption, a well-distributed pore structure was supported by high porosity and a large number of open pores [31].

#### 2.1.2. TEM Study

Figure 3 illustrates the properties of the NLS/Cs and the NLS/Cs/Alg. nanocomposite. The TEM micrograph reveals that the constructed NLS/Cs/Alg. showed a multilayer structure with a crystal size of ≈8 nm. The elements of the NLS composite were spherical and sparsely distributed with spherical crystals of ≈13 nm on the particle surfaces, as shown in figure. This irregular surface indicates that nucleation occurred when the NLS was dropped onto the alginate. In other words, the film covering the particles nucleated on the alginate [32].

### 2.2. Effect of pH

In order to evaluate the effects of pH in the sorption of CR and BG dyes, the tests were carried out in the pH range of 3.0 to 9.0, and 12 mg·L of each dye and 0.02 g·L of NLS/Cs/Alg. nanocomposite were added to a conical flask with stirring for 90 min. The variation of the adsorption capacity of the nanocomposite with pH is shown in Figure 4A. The surface of the nanolimestone was slightly negatively charged according to the zeta potential measurements [33].

The pH optimum for Congo red was pH 6.5~7. The removal efficiency of the chitosan/alginate was significantly higher at pH 6.5 due to the protonation of the -NH_2_ groups of chitosan to produce positively charged -NH_3_^+^ groups of alginates at a lower pH, which would be suitable for anionic dyes such as CR, while the maximum uptake for BG was observed at pH 8. This could be explained by the fact that below the point zero of charge pH _PZC_ value of 7.3 the surface of the NLS/Cs/Alg. was protonated and changed; thus, it became more positively charged and also the functional sites that were mainly carbonates (CO_3_^2−^) were converted to (HCO_3_^−^). This implies that the positively charged surface enhanced the repulsive forces between the NLS/Cs/Alg. surface and the BG dye. Therefore, the electrostatic attraction between the (HCO_3_^−^) active sites and the BG dye was minimal and resulted in low removal. However, increasing the pH of the solution resulted in the surface of the NLS/Cs/Alg. being progressively deprotonated and negatively charged, leading to enhanced electrostatic interactions between the active sites and the BG dye and to higher adsorption [11,34].

### 2.3. Effect of Contact Time

The variation of the amount of adsorbed dye as a function of time is a significant feature of this experiment because it indicates how fast the adsorption is, from 5 to 90 min. The variation of the adsorption capacity of the nanocomposite with time is shown in Figure 4B. The experiments were carried out in a solution of pH 6.5 with 260 mg·L^−1^ of CR dye, and of pH 8 with 140 mg·L^−1^ of BG dye, along with 0.1 g·L^−1^ of NLS/Cs/Alg. nanocomposite. Due to fast adsorption kinetics, the adsorption was initially very rapid and subsequently slow in the later stages. This effect was presumed to be due to the electrostatic effect. If adsorption is slower, it is because the number of adsorption sites decreases. Equilibrium was established for the CR and BG dye concentrations in 60 min.

### 2.4. Sorption Model

Different kinetic models can be used for studying the adsorption phenomenon, and fitting the experimental data with the models permitted the elucidation of the adsorption mechanism. The kinetics parameters for the sorption of CR and BG dyes on the NLS/Cs/Alg. nanocomposite are represented in Table 1. Generally speaking, there are three steps in an adsorption process: (i) Film diffusion, which is the external mass transfer of the adsorbate from the bulk solution to the external surface of the adsorbent; (ii) pore diffusion or intraparticle diffusion, which is the transport of adsorbate particles from the external surface into the pores of the adsorbent medium; and (iii) the sorption itself, i.e., the surface reaction to attach the adsorbate particles to the internal surface of the sorbent [35,36]. The adsorption of solute from a solution into a sorbent involves the mass transfer of adsorbate (film diffusion), surface diffusion, and pore diffusion. Film diffusion is an independent step, whereas surface and pore diffusion may occur simultaneously. The intraparticle diffusion equation (Equation (1)) represented in Figure 6A, as described by Weber and Morris [37], may be applied in the determination of the intraparticle diffusion rate constant, k_d_, and the boundary resistance, C:q_t_ = k_d_ (t^0.5^) + C,(1)
where q_t_ is the amount of adsorbate adsorbed onto the adsorbent at time t (mg/g), k_d_ is the rate constant (mg/g)min^0.5^, and C determines the boundary layer effect (i.e., the higher the value, the larger the film diffusion resistance) and is linked to external mass transfer [38]. Using the plot of q_t_ versus t^0.5^, it is possible to determine k_d_ and C. When intraparticle diffusion alone is the rate-limiting step, then the plot of q_t_ versus t^0.5^ passes through the origin (C = 0). When film diffusion also takes place then the intercept is C > 0, which gives an idea of the thickness of the boundary layer [38]: The larger the intercept, the greater the boundary layer effect, i.e., the film resistance to mass transfer surrounding the adsorbent is greater [39]. The value of the rate constant for the intraparticle diffusion, k_d_, was evaluated as 10.6 (g/g·min^−1^) and 11.1 for the CR and BG dyes, respectively, giving an indication of the mobility of the dyes toward the composite.

Generally speaking, the faster the adsorption of ions, the higher the probability of a layer of ions being formed on the adsorbent surface (boundary layer on the membrane surface) [40]. Kinetic models are divided into two main categories, according to the rate-limiting step in the adsorption process: Some models are based on the fact that the sorption is the rate-limiting step (adsorption reaction models) and these are usually suitable to explain chemisorption; others suppose that the diffusion is the rate-limiting step (adsorption diffusion models) and these are well suited to physisorption processes [33]. Pseudo-first-order (PFO) or pseudo-second-order (PSO) and intraparticle (IP) models are some examples of adsorption reaction models and diffusion reaction models, respectively. In order to study adsorption kinetics, the linear forms are applied and the suitability of any model depends on the degree of linear correlation between the experimental and the predicted values (R^2^) [36]. The PFO and PSO models rest on the main assumption that adsorption only occurs at localized sites and involves no interaction between the adsorbed ions [40]. According to these models, the process rate depends on the availability of adsorption sites on the surface of the adsorbent rather than on the adsorbate concentration in the bulk solution [41]. Also known as the Lagergren model [42], PFO describes the adsorption of solute onto the adsorbent following the first order mechanism. The linear expression of PFO is reported in Equation (2) and represented in Figure 6B:Log (q_e_ − q) − log q_e_ = −K_ads_·t/2.303,(2)
where q_e_ is the amount of dye adsorbed at equilibrium (mg·g^−1^) and K_ads_ is the first-order rate constant for dye adsorption onto the sorbent (min^−1^). In recent years, the pseudo-second-order rate expression has been widely applied to the adsorption of pollutants from aqueous solutions in order to describe chemisorption involving covalent forces and ion exchange between the adsorbent and the adsorbate [41].

The linear expression of PSO is reported in Equation (2) and represented in Figure 5C. The value of k is determined by plotting:t/q_t_ = 1/k^2^q_e_^2^ + t/q_e_,(3)
where k^2^ is the rate constant of the second-order adsorption (g mg^−1^ min^−1^). Similarly, the slope of the plot of t/qt as a function of t was used to determine the second-order rate constant k^2^.

In order to understand the adsorption kinetics, the order of the adsorption process was ascertained by fitting the experimental data into the pseudo-first-order and pseudo-second-order equations. The equilibrium adsorption capacity (q_t_) toward the CR and BG dyes was calculated by UV–Vis spectroscopy at different time intervals (t) taking 260 and 140 mg·L^−1^ of CR and BG dye, respectively, as well as 0.1 g·L^−1^ of NLS/Cs/Alg. nanocomposite. With increasing times (t = 2, 6, 8, 15, and 70 min), the obtained values of q_t_ are plotted in Table 1. The best fitting was obtained for CR (R^2^ = 0.968) and BG (R^2^ = 0.993) for pseudo-second-order adsorption.

The Bangham equation was used to investigate the amount of BG dye that can be introduced into the pores of the nanocomposite [42]:Log log [C_i_/(C_i_ − q m)] = log (K_o_ m/2.303 V) + α log t,(4)
where K_o_ is the proportionality constant and α is the Bangham equation constant. Kinetics parameters for the sorption of CR and BG dyes on the NLS/Cs/Alg. nanocomposite are shown in Table 1. These results showed that the diffusion of dye into composite pores played a role in the adsorption process [43]. The value of α constant indicated that the sorption of dye was favored to be less than 1.

### 2.5. Isotherm Models

#### 2.5.1. Langmuir Isotherm

In order to model the equilibrium data of the composite and different concentrations of CR and BG dyes, a concentration of 0.1 g/L was applied for the application of the isotherm and thermodynamic models. The Langmuir model was proposed for describing the surface of the composite material, as seen in the following equation [44]:C_e_/q_e_ = 1/b Q_max_ + (1/Q_max_) Ce,(5)
where b is the monolayer adsorption capacity related to the sorption heat (L·mg^−1^) and Q_max_ is the maximum adsorption capacity (mg·g^−1^).

#### 2.5.2. Freundlich Isotherm

The Freundlich expression is an empirical equation describing sorption to a heterogeneous surface [45]. The Freundlich adsorption is presented in Equation (6):ln q_e_ = ln K_f_ + 1/n ln C_e_,(6)
where K_f_ (mol1^−n^ Ln g^−1^) represents the sorption capacity when the dye equilibrium concentration is equal to 1 and n represents the degree of dependence of sorption on the equilibrium concentration. Favorable adsorption was demonstrated by the fact that the value of n was greater than unity.

#### 2.5.3. Dubinin–Radushkevich–Kanager Isotherm

The Dubinin–Radushkevich (D–R) isotherm is more general than the Langmuir isotherm. The model was successfully fit with the cohesive model of adsorption, which deals with the Gaussian energy of distribution. The D–R equation is expressed as follows [45]:ln q = ln q_(DR)_ − βε^2^ and(7)
ε = RT ln (1 + 1/C_e_),(8)
where q_(D-R)_ is the theoretical adsorption capacity (mg g^−1^), β is the activity coefficient related to mean sorption energy (mol^2^ kJ^−2^), ε is the Polanyi potential, R is the ideal gas constant (0.008314 KJ K^−1^ mol^−1^), and T is the absolute temperature (K). E (kJ mol^−1^) is defined as the free energy change required for transferring 1 mole of ions from solution to the solid surfaces, which equals:E = 1/(2ß)^1/2^.(9)

It is useful to analyze the kind of sorption reaction. Sorption is governed by chemical ion exchange if the value of E is in the range of 8–16 kJ mol^−1^. In this case, physical forces may affect sorption. Otherwise, molecular diffusion may be at the forefront as E > 16 kJ mol^−1^ [46].

Based on the D–R model simulation in Table 2, the E values were 7.9 and 7.4 kJ mol^−1^ for CR and BG, respectively, in the range of 8–16 kJ mol^−1^, which implies that adsorption can be either physical–chemical or physical.

### 2.6. Thermodynamic Parameters

The thermodynamic adsorption data of CR and BG dyes on the NLS/Cs/Alg. nanocomposite are represented in Table 3. In order to investigate the effect of temperature on the adsorption of CR and BG dyes on the NLS/Cs/Alg. nanocomposite, the distribution coefficient, K_d_, (L·g^−1^) was calculated at temperatures of 288, 298, 313, and 333 K using Equation (10). The enthalpy and entropy change were calculated from the intercept and slope of the plot of log K_d_ against 1/T, respectively [47,48]:ln K_d_ = −∆H_o_/RT + ∆S_o_/R.(10)

The other thermodynamic parameter, Gibbs free energy (∆G_o_), was calculated by:∆G_o_ = −RT ln K_d_,(11)
where R is the universal gas constant (8.314 J mol^−1^ K^−1^) and T is the temperature (K).

The K_d_ value increased with increasing temperature. The heat of adsorption decreased with increasing temperature. From the dataset of the heat transfer, the enthalpy change (∆H_o_) and the entropy change (∆S_o_) can be calculated. The other thermodynamic parameter as ∆G_o_, can be evaluated by Equation (11). ∆H_o_ indicates that the adsorption of CR and BG dyes on the NLS/Cs/Alg. nanocomposite is endothermic. It showed that the adsorption of chromogenic dyes on the nanocomposite is a random reaction. Going forward, a negative value of ∆G_o_ was displayed, which signified that the adsorption of dye is possible and thermodynamically spontaneous. These results are also in good agreement with the domain-reduction reaction of the D–R isotherm. The adsorption performance of CR and BG dyes onto different adsorbents is described in Table 4.

## 3. Experimental Methods

### 3.1. Primary Materials

The primary materials were crushed limestone, which was collected from stones in Mokkatam, Egypt, sodium alginate powder (C_6_H_9_NaO_7_), chitosan low molecular weight (LMW) (C_6_H_11_NO_4_)_n_ (99% A.R grade), hydrochloric acid (HCl; 32%), sodium hydroxide (NaOH) pallets (98.5% A.R grade), and Congo red (C_32_H_22_N_6_Na_2_O_6_S_2_) and brilliant green (C_27_H_34_N_2_O_4_S) dyes (95% A.R grade). All chemicals were obtained from commercial sources (El-Gomhouria Co. For Trading Drugs, Chemicals & Medical Supplies, Egypt). No further purification was used.

### 3.2. Synthesis of Nanolimestone

The nanolimestone was made by mixing the crushed limestone with HCl acid. The aqueous solution of BaCl_2_ was mixed with 1.5 gm of chitosan LMW, which was dissolved in 3% acetic acid and then treated with triphenyl phosphate (TPP) and distilled. The mixture was heated to 90 °C and mixed with 70 g of sodium carbonate. The mixture was kept for 12 h; then it was filtered, washed with water, and finally calcined at 650 °C for 2 h [56].

### 3.3. Synthesis of Nanolimestone/Alginate Composite

Figure 6 shows a flowchart of the synthesis process for the NLS/Cs/Alg. nanocomposite. The triple nanocomposite was prepared by mixing 2% of *w*/*v* of alginate powder and 7% of the nanolimestone in 500 mL deionized water with stirring for 30 min at 80 °C for a homogenous solution. The mixture was then extruded from a prefabricated injector by controlling the opening of the control valve. Finally, the composite beads were immersed in CaCl_2_ solution for 12 h in order to achieve hardened beads. Distilled water was used for washing the beads many times to remove the excess unbounded CaCl_2_ from the adsorbent surface [12].

### 3.4. Structural and Surface Characterization of the NLS/Cs/Alg. Nanocomposite

#### 3.4.1. Analytical Instruments

The final dried powder was crushed for characterization with a scanning electron microscope (SEM). The surface morphology of our adsorbent was analyzed using a electron probe microanalyzer (JXA-840A, JEOL, Tokoy, Japan) attached to an EDAX unit, with an accelerating voltage of 30 kV and a magnification of 10–400,000×. Transmission electron microscopy (JEM 2100, HRTEM, JEOL, Tokoy, Japan) was also used. The functional groups in the NLS/Cs/Alg. nanocomposite and the interfacial modification were analyzed by Fourier transform infrared spectroscopy (FTIR). The FTIR spectrometer was used at frequencies in the 4000–400 cm^–1^ range. The surface area was determined using BEL SORB-max (MICROTRAC, Tokoy, Japan,). To determine the parameters that had an influence on the adsorption process, measurements of the amounts of CR and BG remaining in the solution after adsorption at a wavelength 497 nm for CR and625 nm for BG were taken using an Visible Spectrophotometer (OPTIMA SP-300, Tokoy, Japan).

#### 3.4.2. Dye Adsorption Studies

All batch experiments were carried out at room temperature (25 °C). The dye solutions with concentrations ranging from 10 to 500 mg L^−1^ were dropped by pipette into quick-fit-top bottles containing 0.01 g of NLS/Cs/Alg. nanocomposite sorbent in 100 mL of aqueous solution. The solution pH was measured at 6.5 for CR and 8 for BG—appropriate for most cases of adsorption. The authors tried to examine how long the adsorption process took to complete and so varied the time at fixed periods (1.0, 5.0, 10, 20, 30, 60, and 90 min). Hence, the present data showed that the CR and BG dyes adsorbed on the NLS/Cs/Alg. nanocomposite within a time of 30 to 60 min. The samples were filtered before being analyzed using a spectrophotometer. In the industrial sector, the filtrated adsorbent may be dried to examine its properties and be weighed again to identify whether some of it is lost during the filtration process and it can also be re-used. Changing the pH of the media would affect the attraction force between the charge of the nanocomposite surface and the dye, so the dye could easily be removed from the nanocomposite and the nanocomposite could be used again [57].

#### 3.4.3. Experimental Data Analysis

The percentage adsorption of CR and BG dyes from the solution was calculated from the relationship as follows:% R = (C_i_ − C_f_ )/C_i_ × 100 and
q = [(C_i_ − C_f_)/m]·V.

The adsorption capacity, q, in mg/g and the removal percentage, %R, of the composite toward CR and BG were evaluated by Equations (1) and (2), where C_i_ and C_f_ are the initial and final concentrations of CR and BG in the solution in mg/L, (v) is the volume of the solution in ml, and m is the mass of the NLS/Cs/Alg. in g.

## 4. Conclusions

In this work, an experimental study on the utilization of an NLS/Cs/Alg. nanocomposite for the removal of Congo red and brilliant green dyes from an aqueous solution was carried out. The following conclusions were made based on the experimental results of the present study. The nanocomposite was characterized by Fourier transform infrared spectroscopy (FTIR), TEM, SEM, energy dispersive X-ray analysis (EDX), and Brunauer–Emmett–Teller BET methods. The potential of this adsorbent was studied for the decolonization of CR and BG; the influence of the initial pH, dye concentration, contact time, adsorbent dose, and temperature on the adsorption of CR and BG was investigated. The adsorption capacity of CR and BG increased with increasing initial dye concentration, while the optimized pH was found to be 7.2 for both. The kinetics of CR and BG removal indicated an optimum contact time of 60 min via a two-stage adsorption kinetic profile (initial fast and subsequent slow equilibrium). The CR and BG adsorption onto NLS/Cs/Alg. follows a pseudo-second-order kinetic model with a determination coefficient (R^2^) very close to unity (~0.983). This relies on the assumption that the chemisorption may be the rate-limiting step, as well as where the CR and BG ions are attached to the adsorbent surface by forming chemical bonds, tending to find sites that maximize their coordination number with the surface. The equilibrium adsorption data for BG on NLS/Cs/Alg. were analyzed by various models. The results indicate that the Langmuir isotherm provides the best correlation (Q_max_ ~3890 mg/g at 299 K for CR and ~3080 mg/g for BG). The negative free enthalpy, ΔG, and positive enthalpy, ΔH, indicated that the adsorption of CR and BG onto NLS/Cs/Alg. was spontaneous and endothermic over the studied temperature range. CR and BG were strongly bonded to the adsorbent surface, while the positive entropy, ΔS, stated clearly that the randomness increased at the solid–solution interface during dye adsorption, and that there was some structural exchange between the active sites and the dye ions. The comparison of the adsorption capacity of our adsorbent with others showed its attractive properties from both industrial and economic perspectives. This study produced encouraging results.

## Figures and Tables

**Figure 1 molecules-26-02586-f001:**
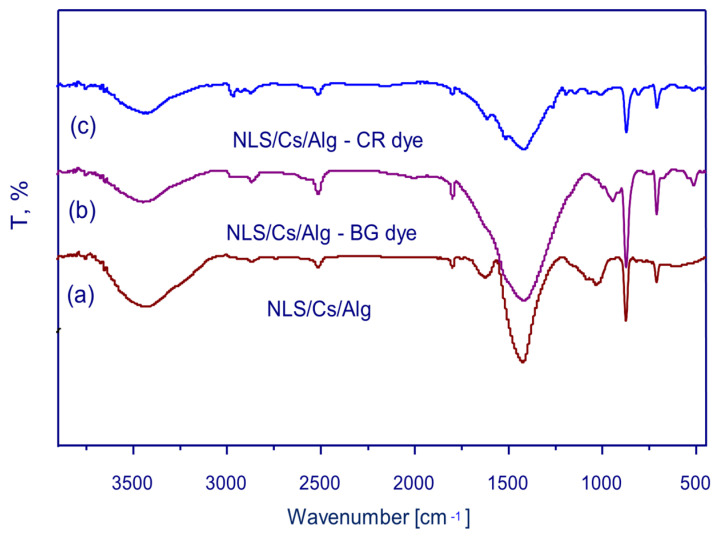
FTIR for (**a**) limestone–chitosan–alginate (NLS/Cs/Alg.), (**b**) NLS/Cs/Alg. brilliant green (BG) dye, and (**c**) NLS/Cs/Alg. Congo red (CR) dye.

**Figure 2 molecules-26-02586-f002:**
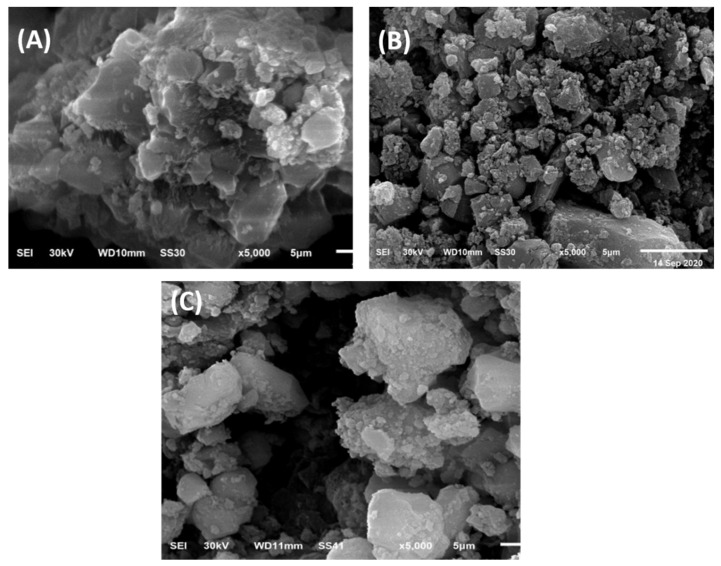
SEM of (**A**) NLS/Cs/Alg., (**B**) NLS/Cs/Alg.–CR dye, and (**C**) NLS/Cs/Alg.–BG dye.

**Figure 3 molecules-26-02586-f003:**
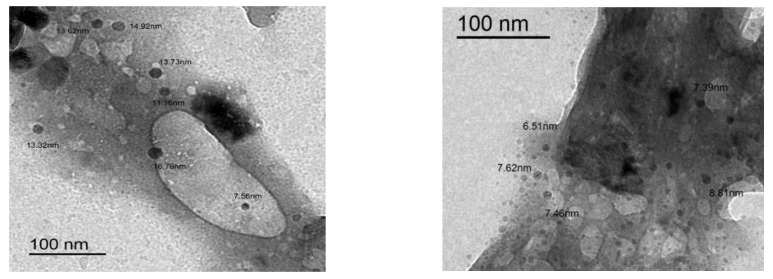
TEM micrographs of the synthesized NLS/Cs/Alg. composite.

**Figure 4 molecules-26-02586-f004:**
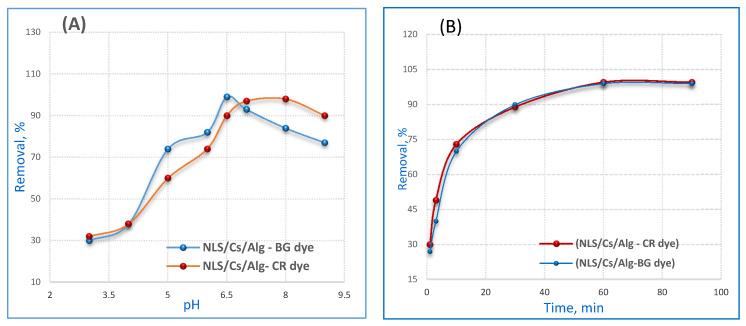
Influence of (**A**) pH and (**B**) time on the adsorption of 12 mg/L CR or BG dye by 0.02 g/100 mL NLS/Cs/Alg. nanocomposite.

**Figure 5 molecules-26-02586-f005:**
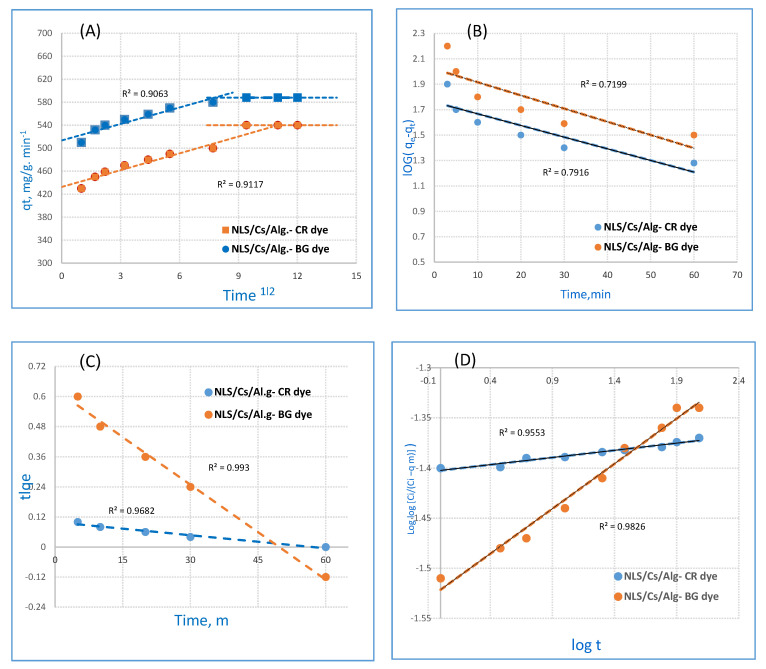
Nanocomposite: (**A**) Morris–Weber equation, (**B**) pseudo–first–order (PFO), (**C**) pseudo–second–order (PSO), and (**D**) Bangham.

**Figure 6 molecules-26-02586-f006:**
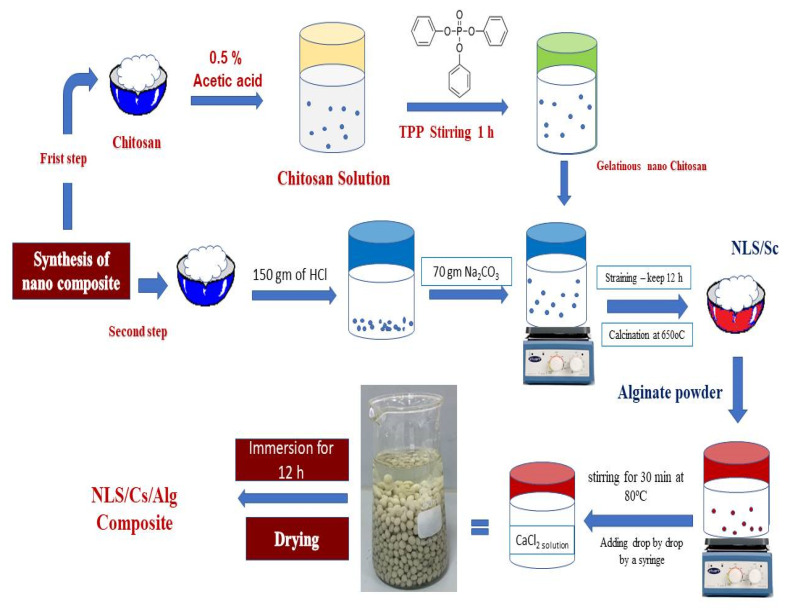
Flowchart of the NLS/Cs/Alg. formation.

**Table 1 molecules-26-02586-t001:** Kinetics parameters for the sorption of CR and BG dyes on the NLS/Cs/Alg. nanocomposite.

Lagrange (Pseudo-First-Order)	K_ads_ (min^−1^)	q_e_, cal (mg g^−1^)	R^2^
CR dye	0.012	94.9	0.719
BG dye	0.018	31.7	0.791
(Pseudo-Second-Order)	K_2_ (g mg^−1^ min^−1^)	q_e_, cal (mg g^−1^)	R^2^
CR dye	2.07	94.9	0.968
BG dye	1.99	31.7	0.993
Bangham	A	K_o_	R^2^
CR dye	0.04	87.7	0.982
BG dye	0.7	128.9	0.955
Morris–Weber	K_d_ (mg g^−1^ min^0.5^)	C (mg·g^−1^)	R^2^
CR dye	10.7	510	0.911
BG dye	11.1	430	0.906

**Table 2 molecules-26-02586-t002:** Calculated parameters of the Langmuir, Freundlich, and Dubinin–Radushkevich (D–R) models for the adsorption of CR and BG dyes on the NLS/Cs/Alg. nanocomposite.

**Langmuir**			
Dye	b (L·mg^−1^)	Q_max_ (mg·g^−1^)	R^2^
CR	5.3 × 10^−3^	2250	0.998
BG	6.6 × 10^−3^	2010	0.988
**Freundlich**			
Dye	K_f_ (moln^−1^ Ln g^−1^)	n	R^2^
CR	312	3.3	0.998
BG	178	2.1	0.979
**D–R model**			
Dye	E (kJ mol^−1^)	q_(DR)_ (mg·g^−1^)	R^2^
CR	7.9	3890	0.995
BG	7.4	3080	0.968

**Table 3 molecules-26-02586-t003:** Thermodynamic data for the adsorption of CR and BG dyes on the NLS/Cs/Alg. nanocomposite.

∆S_o_ (J·mol^−1^·K^−1^)	∆H_o_ (kJ·mol^−1^)	∆G_o_ (kJ·mol^−1^)	ln K_d_	T(K)	Dye
89.1	37.7	−5.1	10.49	288	CR
		−27.1	10.82	298	
		−29.9	11.41	313	
		−32.4	11.72	323	
79.1	8.71	−21.5	9.02	288	BG
		−22.5	9.14	298	
		−24.8	9.44	313	
		−27.4	9.70	323	

**Table 4 molecules-26-02586-t004:** Adsorption performance of CR and BG dyes onto different adsorbents.

Dye	Adsorbent	q_e_ (mg/g)	References
CR	Spherical Fe_3_O_4_ nanoparticles functionalized with 1, 2, 4, 5-benzenetetracarboxylic acid	192	[49]
	Synthesis of siderite	9416	[50]
	Iron nanoparticles	887	[51]
	Shiitake mushroom	217.8	[52]
BG	Activated carbon	90	[53]
	Activated carbon derived from medlar nucleus (ACMN)	833.33	[54]
	Nanohydroxyapatite	49.1	[18]
	Low cost agricultural waste	24.23	[55]

## Data Availability

Date of the compounds are available from the authors.

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
