# Peer review of "Equilibrium and Kinetic Study of Anionic and Cationic Pollutants Remediation by Limestone–Chitosan–Alginate Nanocomposite from Aqueous Solution"

_molecules, 2021, doi:10.3390/molecules26092586_

Round 1
Reviewer 1 Report
the manuscript could be accepted in the presebt form after the modifciations apported by the authors
Author Response
Point 1 : English language and style are fine/minor spell check required.
Respond 1: The manuscript has been sent for English Editing

Reviewer 2 Report
This manuscript (molecules-1174011) reported the preparation of limestone-chitosan-alginate nanocomposites for the removal of brilliant green and Congo red dyes in aqueous solutions. The effect of contact time, temperature, concentration, and pH on the adsorption performance of nanocomposites were investigated in detail. The preparation of limestone-chitosan-alginate nanocomposites with interconnected pores are novel, and this materials shows high adsorption capacity for brilliant green and Congo red. However, there are still some issues to be addressed. Therefore, a major revision is suggested before its acceptance.
- Macropore size distributions, nitrogen adsorption/desorption isotherms,BJH mesopore size distribution and BET specific surface area of limestone-chitosan-alginate nanocomposites should be added to characterize the pore structure of the nanocomposites.
- The persuasive explanation of adsorption mechanism for dyes should be added.
- “2.1.1 SEM stud”should be corrected into “2.1.2 SEM study”.
- The English (grammar and syntax) should be improved carefully.
Author Response
Dear Reviewer
I hope this mail find you well
All the responds for your comments have been added at the end of the manuscript
Beast Regards

Round 2
Reviewer 2 Report
The reviewer agrees with the revision, and the revised manuscript can be accepted.
This manuscript is a resubmission of an earlier submission. The following is a list of the peer review reports and author responses from that submission.